# Broad-spectrum infrared thermography for detection of M2 digital dermatitis lesions on hind feet of standing dairy cattle

**Arne Vanhoudt**[1¤a]*, **Casey Jacobs**[2¤b], **Maaike Caron**[1¤c], **Herman W. Barkema**[2], **Mirjam Nielen**[1], **Tine van Werven**[1,3], **Karin Orsel**[2]

**1** Department of Population Health Sciences, Faculty of Veterinary Medicine, Utrecht University, Utrecht, Utrecht, The Netherlands, **2** Department of Production Animal Health, Faculty of Veterinary Medicine, University of Calgary, Calgary, Alberta, Canada, **3** University Farm Animal Practice, Utrecht University, Harmelen, Utrecht, The Netherlands

¤a Current address: Royal GD, Deventer, Overijssel, The Netherlands
¤b Current address: More Than Just Feed, Alberta, Canada
¤c Current address: Vee&Arts Farm Animal Practice, Bergen op Zoom, North Brabant, The Netherlands
* vanhoudt.arne@gmail.com

**Data Availability Statement:** Due to a signed agreement between the researchers and farmers we are unable to make all data underlying the findings of this manuscript fully available without

## Abstract

Low-effort, reliable diagnostics of digital dermatitis (**DD**) are needed, especially for lesions warranting treatment, regardless of milking system or hygienic condition of the feet. The primary aim of this study was to test the association of infrared thermography (**IRT**) from unwashed hind feet with painful M2 lesions under farm conditions, with lesion detection as ultimate goal. Secondary objectives were to determine the association between IRT from washed feet and M2 lesions, and between IRT from unwashed and washed feet and the presence of any DD lesion. A total of 641 hind feet were given an M-score and IRT images of the plantar pastern were captured. Multivariable logistic regression analyses were done with DD status as dependent variable and maximum infrared temperature (**IRTmax**), lower leg cleanliness score and locomotion score as independent variables, and farm as fixed effect. To further our understanding of IRTmax within DD status, we divided IRTmax into two groups over the median value of IRTmax in the datasets of unwashed and washed feet, respectively, and repeated the multivariable logistic regression analyses. Higher IRTmax from unwashed hind feet were associated with M2 lesions or DD lesions, in comparison with feet without an M2 lesion or without DD, adjusted odds ratio 1.6 (95% CI 1.2–2.2) and 1.1 (95% CI 1.1–1.2), respectively. Washing of the feet resulted in similar associations. Dichotomization of IRTmax substantially enlarged the 95% CI for the association with feet with M2 lesions indicating that the association becomes less reliable. This makes it unlikely that IRTmax alone can be used for automated detection of feet with an M2 lesion. However, IRTmax can have a role in identifying feet at-risk for compromised foot health that need further examination, and could therefore function as a tool aiding in the automated monitoring of foot health on dairy herds.

restriction. Restrictions on this dataset were imposed by Cattle Health Research Group, Department of Production Animal Health, Faculty of Veterinary Medicine, University of Calgary, Calgary, Alberta, Canada. Access to the anonymised dataset for research involving the research group that collected the original data is possible, and enquiries for this purpose can be made by contacting the corresponding author or research group (headpah@ucalgary.ca or karin.orsel@ucalgary. ca).

**Funding:** This study was funded by CAAP (Canadian Agriculture Adaptation Program; Canada; http://omaf.gov.on.ca/english/food/ industry/can-agri-adapt.htm), Alberta Milk (Edmonton, Alberta, Canada; https://albertamilk. com/), and DeLaval Manufacturing (Kansas City, Missouri, United States of America; https://www. delaval.com/en-us/) The funders had no role in study design, data collection and analysis, decision to publish, or preparation of the manuscript.

**Competing interests:** The authors have declared that no competing interests exist.

## Introduction

Digital dermatitis (**DD**) is a multifactorial, infectious, polytreponemal disease, characterized by ulcerative or hyperkeratotic lesions that are typically located between the heel bulbs of hind feet [1]. It affects dairy cattle worldwide and cattle with DD have reduced animal welfare, production and reproductive performance, resulting in economic losses and increased labour for the farmers [2–4].

Current control of DD relies on keeping the disease in a manageable state [5] and entails both disease prevention through footbathing at herd level and treatment of ulcerative lesions at cow level. These ulcerative lesions are commonly grouped as active lesions and consist of the M1, M2, and M4.1 stage lesions [6].

Detection of DD lesions is often late and typically takes place either during routine foot trimming or when cows are seen lame or standing on tiptoes due to a painful lesion. Visual inspection of the feet in the trimming chute is considered best practice for the diagnosis of DD [7]. However, often this is not practical due to time and labour requirements and typically is not performed on a routine basis at herd level which is essential for early detection and treatment of M2 lesions [8]. Prompt effective treatment of M2 lesions deals with the welfare aspect of DD, as Higginson Cutler et al. [3] described these lesions as most painful.

Consequently, scoring feet in the milking parlour after feet have been hosed off with water was successfully tested as an alternate diagnostic tool, compared to identification in the trimming chute, with a sensitivity (**Se**) and specificity (**Sp**) for detecting M2 lesions of about 0.60 and 1.00, respectively [7]. Others compared scoring in the milking parlour with the trimming chute for presence or absence of a DD lesion, regardless the M-stage, and reported Se 0.55–1.00 and Sp 0.80–1.00 [8–10]. Due to the absence of a milking parlour on dairy herds with an automatic milking system, routine screening of DD on these herds must occur during pen walks or by running the entire herd through the trimming chute. Cramer et al. [8] reported pen walks to have poor discerning capacity for M-stages of DD.

There is, therefore, an urgent need for a reliable method to quickly and easily diagnose M2 lesions which is widely applicable regardless the hygienic condition of the feet, nor dependent of milking system. A small number of studies investigated the use of infrared thermography (**IRT**) for the purpose of detecting the presence of DD, regardless the M-stage. This technology is based on detecting infrared radiation, which is emitted by all objects, depending on their temperature. Skin temperature is highly dependent on the temperature of the underlying tissue and circulation. Therefore, variations in skin temperature, captured by an IRT camera, can be related to underlying inflamed tissue or altered metabolic activity [11], as may occur during inflammation caused by DD. In a study by Stokes et al. [12], maximum infrared temperature (**IRTmax**) of the plantar pastern was higher on feet with DD from standing cattle in comparison with feet without any lesions. However, IRTmax was not different between feet with DD lesions and feet with other lesions [12]. Alsaaod et al. [13] were able to detect hind feet with DD in standing cows using the difference between IRTmax of hind and front feet.

For practical and technical reasons, M2 detection on unwashed feet is preferred over detection on pre-washed feet [12]. The primary objective of this study was, therefore, to determine whether broad spectrum IRT from unwashed hind feet of cows standing in a milking parlour was associated with M2 lesions. As secondary objectives, we investigated the association of IRT from pre-washed standing hind feet with M2 lesions and the association of IRT from unwashed and washed standing hind feet with the presence of DD, regardless of M-score.

## Materials and methods

### Study design and ethical statement

We analysed data collected in parallel with the published randomized controlled trial by Jacobs et al. [14]. The IRT measurements and locomotion scores (**LS**) were not analysed before, whereas the M-scores, lower leg cleanliness scores (**CS**), and farm descriptives were used from Jacobs et al. [14]. All methods were approved by the Animal Care Committee (AC13-0055) of the University of Calgary. Written informed consent was obtained from the herd owners prior to participation in the study.

Participating dairy farms met the following criteria: $\geq$ 60 lactating dairy cows, > 90% Holstein-Friesian cows, lactating cows housed in freestall barns and milked in a parlour. On a convenience sample of four farms, a target of 40 dairy cows were semi-randomly selected by dividing the number of milking cows, as stated by the farmer, by 40 and selecting every $n^{th}$ cow in the milking parlour. These four farms were visited at 3-week intervals for a total of 12 weeks, resulting in five visits with data collection per farm. An opportunistically selected fifth farm, was visited once to collect IRT images and M-scores only. On this fifth farm, data was collected from as many hind feet as possible without delaying the milking routine. This resulted in data collection from 131 of the 186 cows being milked during the visit. Each farm was located in Alberta, Canada, and data were collected from May to August 2013 on the first four farms and in November 2013 on the fifth farm. The routine treatment and hoof trimming schedule was maintained for all farms over the course of the study [14]. We refer the reader to Jacobs et al. [14] for details on the footbathing practices for lactating cows. Where farm 1 corresponds with farm C4, farm 2 with farm C3, farm 3 with Q3, farm 4 with Q4, and farm 5 with Q6 in Jacobs et al. [14].

### Clinical scores and infrared thermography data collection

One observer (CJ), trained in scoring using digital colour images, videos and definitions, scored all feet in the study and took all IRT images. During data collection the observer was aided by one other person to keep records. All data were collected during milking from both standing hind feet of recruited cows only. First, data were collected from recruited cows on one side of the parlour, followed by recruited cows on the other side of the parlour. The order of data collection remained the same throughout the study: CS, IRT image capture of unwashed feet, wash feet with water using a water source that was available in the parlour, IRT image capture of washed feet, and M-score washed feet. For the IRT images of washed feet, the amount of time between washing feet and capturing the second IRT image varied according to the milking routine and size of the milking parlour. Recruited cows were video recorded while exiting the milking parlour, and these recordings were used for locomotion scoring.

The CS was done as developed by Cook [15] and adapted by Solano et al. [16] and was scored from 1 to 4 according to varying contamination: 1 = fresh manure for < 50%; 2 = fresh manure for > 50%; 3 = dried caked and fresh manure for > 50%; and 4 = entire area with dried caked manure. Scoring for DD was according to the M-stage classification developed by Döpfer et al. [17], using a headlamp and a cosmetic mirror glued to a kitchen spatula [7, 9]. In summary, M0 was defined as normal digital skin with no evidence of DD; M1 was defined as a small (< 2 cm in diameter) circumscribed red to grey epithelial defect; M2 was defined as an ulcerative lesion $\geq$ 2 cm in diameter with a red-grey surface; M3 was defined as a stage characterized by a firm dark scab-like covering; and M4 was characterized by a lesion surface with brown or black tissue that was hyperkeratotic, scaly, or proliferative. As in Jacobs et al. [14], the M4.1 lesions, with small red circumscribed lesions occurring within the boundaries of an

existing M4 lesion [18], were not scored as such, and therefore lesions of this description were included within the M1 category. The LS considered five classes, with 1 = perfect gait and 5 = severely lame, based on the 7 specific gait attributes as described by Flower and Weary [19] and validated for use on video recordings by Chapinal et al. [20] and Ito et al. [21].

## Infrared thermography imaging

Thermal images of all hind feet enrolled in the study were obtained with a FLiR i3 handheld thermal imaging camera (FLiR Systems Inc.) and analysed using ThermaCAM Researcher Professional 2.8 SR-2 software (FLiR Systems Inc.). Details on the technical characteristics of the camera are provided in S1 Table. The software package produced specific information such as minimum, maximum and mean temperature with standard deviation for whole images or within a specific area using a geometric figure drawn on the image. Thermal images of the plantar pastern, focused on the cleft between the heel bulbs, were taken at a distance of approximately 0.5 m. To analyse the IRT images, the rectangle tool of the software was used to select the plantar aspect of the hind feet from the bottom of the dewclaws to the heel (Fig 1). The processing of all IRT images in the software, including the drawing of the rectangles, was done by one observer (MC). Previous studies identified IRTmax as the most suitable IRT variable for research on the association between IRT and foot health [12, 22], hence we only used IRTmax for the analyses in our study. Thermograph resolution was calibrated to ambient temperature before each collection session using a Reed LM-800 4-in-1 pocket thermo-anemometer, hygrometer, thermometer and illuminometer (Reed Instruments).

## Statistical analyses

Statistical analyses were conducted using RStudio Version 1.3.1093 [23, 24]. Statistical significance was declared at p < 0.05. Handling of the collected data for analysis of the different objectives is detailed in Fig 2.

First, descriptive analyses were done to identify the number of feet with M2 lesions with IRTmax available after software processing of the IRT images. At the first visit, 21 hind feet met these requirements in the unwashed and washed condition. Another 15 unwashed and 19

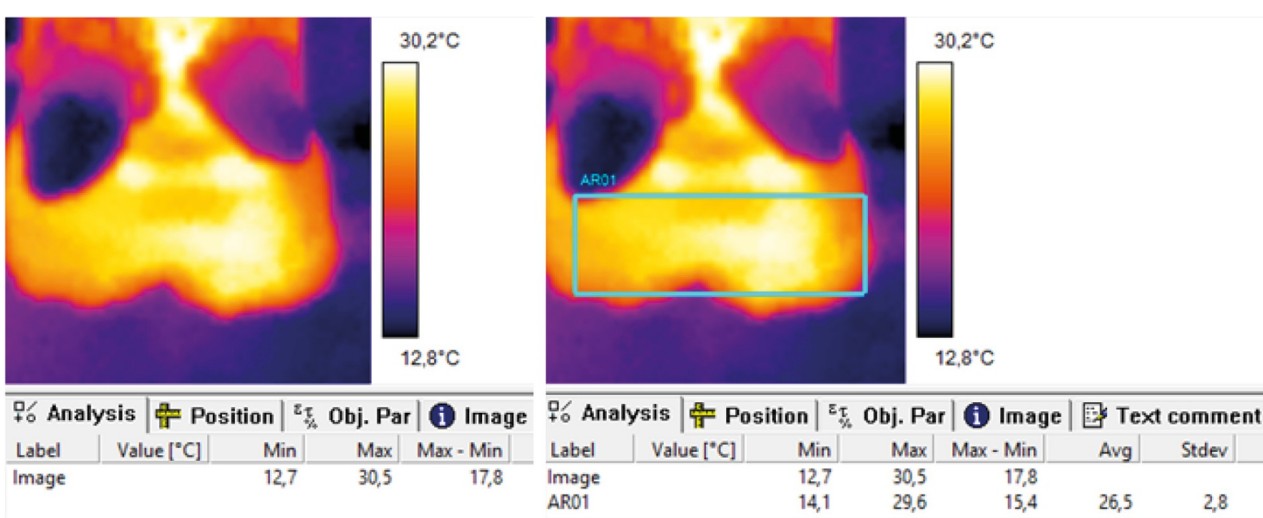

**Fig 1. Example of infrared thermography data collection and analysis of images from FLiR i3 handheld camera using ThermaCAM Researcher Professional 2.8 SR-2 software.**

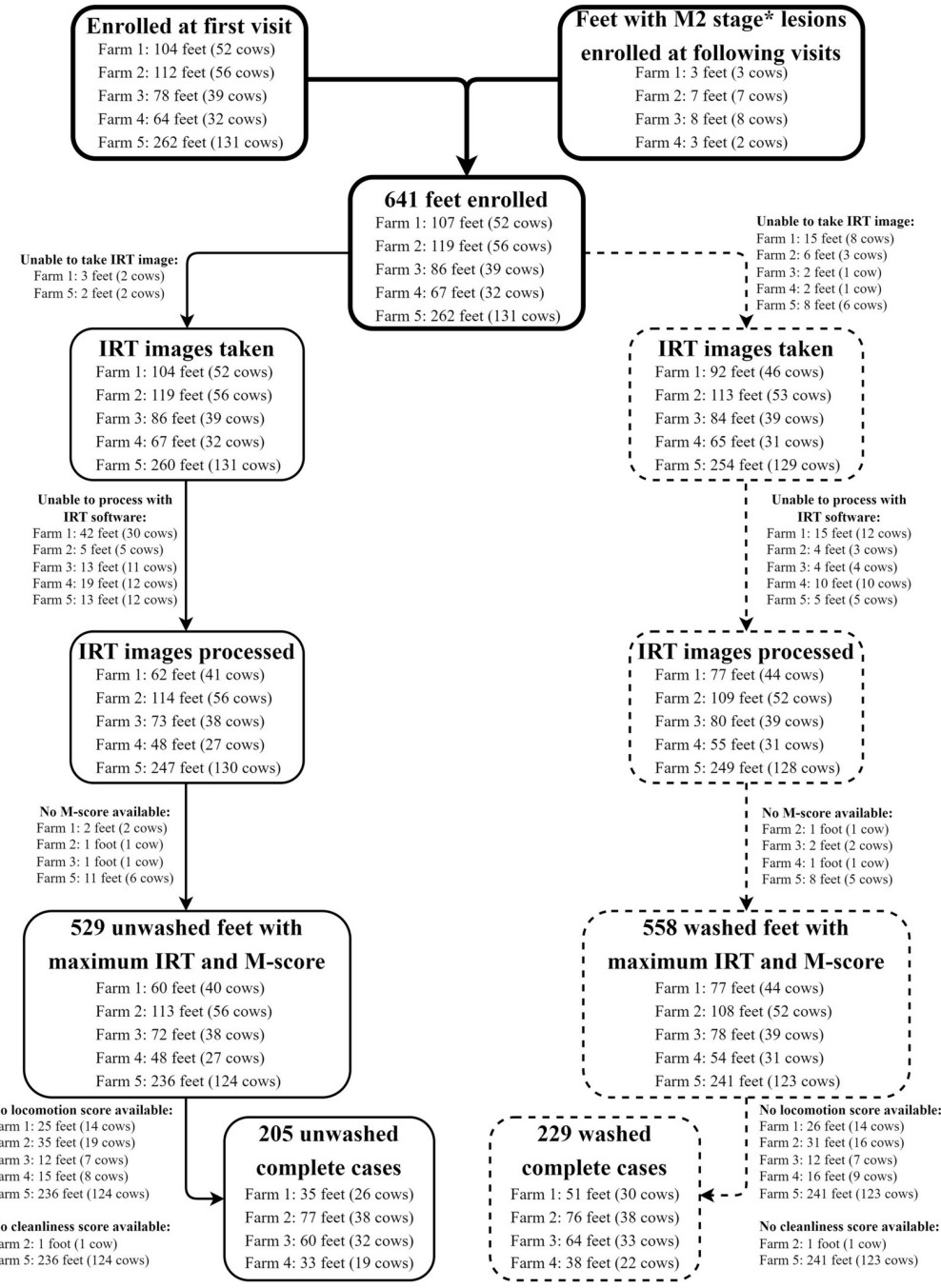

**Fig 2. Study flow diagram for a study testing the association between broad spectrum infrared thermography and the presence of digital dermatitis lesions using unwashed and washed hind feet from five Canadian dairy herds.** The first four herds were visited at 3-week intervals for a total of 12 weeks, resulting in five visits with data collection per farm and the fifth farm was visited once. The bold lines represent general study recruitment, the solid lines represent the unwashed hind feet and the dotted lines represent the washed hind feet. The diagram was created with www.app.diagrams.net.

washed hind feet were available from the other visits. Because of the low prevalence of M2 lesions in the dataset, it was decided to complement the data from the first visit with M2 scored feet only from the following visits for further statistical analyses.

Prior to statistical analyses, CS and LS were dichotomized. Dichotomization of the CS was based on presence of dried manure or not, with CS 1 and 2 categorized as 'fresh manure' and CS 3 and 4 as 'dried manure' [25]. Dichotomization of the LS was based on presence of limping indicating lameness with LS 1 and 2 as 'not lame' and LS 3, 4, and 5 as 'lame' [26, 27]. Associations were first assessed using univariable logistic regression analyses between DD status and IRTmax, CS, LS, and farm, respectively; and second using multivariable logistic regression analysis. The dependent variable was DD status (M2 = 1 and M0|M1|M3|M4 = 0; or DD present = 1 and absent = 0) and independent variables were IRTmax, CS, and LS. Farm was fixed into the model as a means to account for farm effect and clustering of cows within farm. The final reduced model was based on the lowest Akaike information criterion using a backward elimination approach [28]. Univariable logistic regression analyses used both the full categorical and dichotomized CS and LS, and results hereof informed variable selection for the multivariable models. To further our understanding of IRTmax within DD status, we divided IRTmax into two groups over the median value of IRTmax, regardless of M-score, in the datasets of unwashed and washed feet, respectively, and repeated the multivariable logistic regression analyses as described above. The full results of all regression analyses are reported in the supporting information (S1 File).

## Results

A total of 641 hind feet from 310 cows of 5 farms were enrolled in the study (Fig 2). After discarding feet missing an IRTmax value, either due to absence of an IRT image or inability to process the IRT image with the software, and discarding feet missing an M-score, a total of 529 unwashed hind feet from 285 cows and a total of 558 washed hind feet from 289 cows with an IRTmax value and an M-score were available for analysis. The unwashed dataset had 54 cows with one observation, 218 cows with two observations, and 13 cows with three observations with IRTmax and M-score data, whereas the washed dataset had 32 cows with one observation, 245 cows with two observations, and 12 cows with three observations with IRTmax and M-score data. From these, 205 unwashed hind feet from 115 cows and 229 washed hind feet from 123 cows also had both LS and CS data available.

Lactating herd size ranged from 166 to 279 cows and farm-level DD prevalence (at least one hind foot with DD) in enrolled cows ranged from 62 to 85% (mean 72%, standard deviation 9) at the first visit. An overview of the M-scores by farm, LS, and CS for the hind feet with an IRTmax in our study is provided in Table 1. The unwashed hind feet dataset contained 36 feet with an M2 lesion and 493 feet without an M2 lesion, and 310 feet with DD and 219 feet without DD. The washed hind feet dataset contained 40 feet with an M2 lesion and 518 feet without an M2 lesion, and 329 feet with DD and 229 feet without DD. Table 2 provides an overview of the descriptive statistics of IRTmax for each group of hind feet and boxplots of the IRTmax are provided in Fig 3.

### Association of maximum infrared temperature with the presence of M2 lesions

In the final multivariable logistic regression analysis models of our study, higher IRTmax values were associated with an increased odds for M2 lesions on both unwashed (adjusted OR 1.6; 95% CI 1.2–2.2) and washed hind feet (adjusted OR 1.4; 95% CI 1.1–1.7), as was presence of dried manure on the lower hind legs (CS = 3 and 4; Table 3). These associations remained similar after dichotomization of IRTmax with an adjusted OR of 13.9 (95% CI 3.4–95.7) and 4.8 (95% CI 1.7–15.8) for unwashed and washed hind feet, respectively.

**Table 1. M-scores for digital dermatitis, locomotion score, and cleanliness score for hind feet before and after washing from five Canadian dairy herds.**

| | Unwashed hind feet | | | | | | Washed hind feet | | | | | |
|---|---|---|---|---|---|---|---|---|---|---|---|---|
| | M0 | M1 | M2 | M3 | M4 | Total | M0 | M1 | M2 | M3 | M4 | Total |
| **Herd** | | | | | | | | | | | | |
| **1** | 22 | 0 | 2 | 21 | 15 | 60 | 32 | 0 | 3 | 24 | 18 | 77 |
| **2** | 27 | 1 | 14 | 50 | 21 | 113 | 23 | 1 | 13 | 51 | 20 | 108 |
| **3** | 36 | 1 | 7 | 13 | 15 | 72 | 36 | 1 | 9 | 15 | 17 | 78 |
| **4** | 15 | 0 | 1 | 28 | 4 | 48 | 14 | 0 | 3 | 32 | 5 | 54 |
| **5** | 119 | 2 | 12 | 33 | 70 | 236 | 124 | 2 | 12 | 34 | 69 | 241 |
| **Total** | 219 | 4 | 36 | 145 | 125 | 529 | 229 | 4 | 40 | 156 | 129 | 558 |
| **LS[b]** | | | | | | | | | | | | |
| **1** | 45 | 1 | 10 | 57 | 20 | 133 | 49 | 1 | 12 | 60 | 23 | 145 |
| **2** | 10 | 0 | 6 | 20 | 7 | 43 | 11 | 0 | 6 | 19 | 8 | 44 |
| **3** | 7 | 0 | 0 | 8 | 6 | 21 | 11 | 0 | 2 | 11 | 8 | 32 |
| **4** | 4 | 0 | 2 | 2 | 0 | 8 | 4 | 0 | 2 | 3 | 0 | 9 |
| **5** | 0 | 0 | 0 | 0 | 1 | 1 | 0 | 0 | 0 | 0 | 0 | 0 |
| **Total** | 66 | 1 | 18 | 87 | 34 | 206 | 75 | 1 | 22 | 93 | 39 | 230 |
| **CS[c]** | | | | | | | | | | | | |
| **1** | 10 | 0 | 2 | 8 | 6 | 26 | 6 | 0 | 2 | 10 | 6 | 24 |
| **2** | 61 | 2 | 7 | 66 | 29 | 165 | 65 | 2 | 6 | 73 | 32 | 178 |
| **3** | 26 | 0 | 14 | 34 | 20 | 94 | 32 | 0 | 19 | 36 | 22 | 109 |
| **4** | 3 | 0 | 0 | 4 | 0 | 7 | 2 | 0 | 0 | 3 | 0 | 5 |
| **Total** | 100 | 2 | 23 | 112 | 55 | 292 | 105 | 2 | 27 | 122 | 60 | 316 |

[a] M-stages [17] were determined in-parlour, after washing the feet with water. The M4.1 stage by Berry et al. [18] is included in the M1 stage.

[b] Locomotion scores [19] were determined from video recordings of cows exiting the milking parlour with score $\geq$ 3 considered lame; only available for feet from farm 1 to 4.

[c] Lower leg cleanliness scores [15, 16] were determined in-parlour with presence of dried manure in score $\geq$ 3; only available for feet from farm 1 to 4.

## Association of maximum infrared temperature with the presence of digital dermatitis lesions

Multivariable logistic regression analyses identified that higher IRTmax values were associated with an increased odds for DD presence on both unwashed (adjusted OR 1.1; 95% CI 1.1–1.2)

**Table 2. Descriptive statistics for maximum infrared temperature (˚C) of the plantar pastern from standing dairy cattle feet before and after washing, categorized by digital dermatitis (DD) status.**

| DD status | N | mean | SD | minimum | Q1 | median | Q3 | maximum |
|---|---|---|---|---|---|---|---|---|
| **Unwashed hind feet** | | | | | | | | |
| **M2[a]** | 36 | 32.1 | 1.2 | 28.8 | 31.5 | 32.2 | 33.0 | 34.3 |
| **M0\|M1\|M3\|M4[a]** | 493 | 30.3 | 2.8 | 18.4 | 29.0 | 30.7 | 32.2 | 35.4 |
| **DD present** | 310 | 30.9 | 2.4 | 20.3 | 29.7 | 31.2 | 32.5 | 35.4 |
| **DD absent** | 219 | 29.7 | 3.1 | 18.4 | 28.1 | 30.3 | 31.9 | 34.4 |
| **Washed hind feet** | | | | | | | | |
| **M2[a]** | 40 | 32.1 | 1.3 | 29.0 | 31.3 | 32.3 | 32.9 | 34.7 |
| **M0\|M1\|M3\|M4[a]** | 518 | 30.5 | 3.0 | 17.3 | 29.3 | 31.2 | 32.5 | 35.3 |
| **DD present** | 329 | 31.1 | 2.4 | 19.0 | 29.8 | 31.5 | 32.8 | 35.3 |
| **DD absent** | 229 | 29.9 | 3.4 | 17.3 | 28.4 | 31.0 | 32.4 | 35.1 |

[a] M-stages [17] were determined in-parlour, after washing the feet with water; the M4.1 stage by Berry et al. [18] is included in the M1 stage.

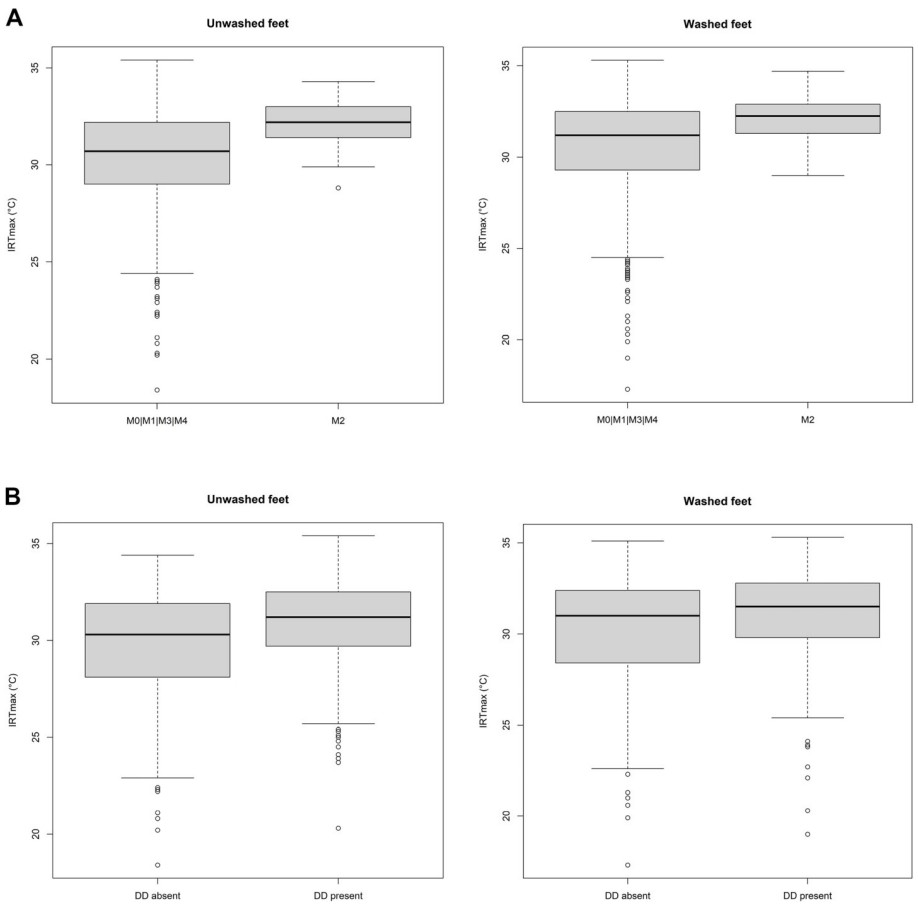

**Fig 3. Boxplots for maximum infrared temperature (IRTmax) from the pastern region of dairy cattle hind feet before and after washing.** (A) For hind feet with M2 or M0|M1|M3|M4 lesions of digital dermatitis (DD). (B) For hind feet with absence or presence of DD. Bold solid line = median, box = interquartile range (IQR), bottom whisker = $25^{th}$ percentile—1.5 x IQR, top whisker = $75^{th}$ percentile + 1.5 x IQR, circle = datapoint outside the interwhisker range.

and washed hind feet (adjusted OR 1.1; 95% CI 1.1–1.2; Table 4). This association disappeared after dichotomization of IRTmax.

## Discussion

This multi-farm study provides insights into the practical application of IRT for detection of DD, M2 lesions in particular, on hind feet from standing cows with a handheld IRT camera. Higher IRTmax values were associated with feet with M2 lesions or DD lesions, in comparison with feet without an M2 lesion or without DD, respectively, regardless of the hygienic condition of the feet. Dichotomization of IRTmax substantially enlarged the 95% CI for the association with feet with M2 lesions indicating that the association becomes less reliable. When looking at feet with any DD lesions, there was no association with the dichotomized IRTmax. Previous work reported poor test characteristics to diagnose the presence of DD lesions using IRTmax with Se 0.75–0.89 and Sp 0.65–0.70 [13, 29]. Altogether, these findings suggest that it is unlikely that a cut-off value for IRTmax with high Se and Sp for the detection of feet with M2 lesions can be determined using cross-sectional data.

**Table 3. Final reduced multivariable logistic regression models to test the association between maximum infrared temperature (IRTmax) from the plantar pastern and presence of M2 lesions [17] of digital dermatitis on hind feet from standing dairy cattle before and after washing, with lower leg cleanliness score (CS) [15, 16] as explanatory variable and farm as fixed effect.**

| Model | Variable | Adjusted OR | 95% CI |
|---|---|---|---|
| **Unwashed hind feet** | | | |
| **Continuous IRTmax** | | | |
| **IRTmax + CS + farm** | IRTmax | 1.6 | 1.2–2.2 |
| | CS fresh manure | 1 | referent |
| | CS dried manure | 4.1 | 1.6–10.7 |
| | Farm 1 | 1 | referent |
| | Farm 2 | 2.1 | 0.5–14.2 |
| | Farm 3 | 5.5 | 1.2–40.5 |
| | Farm 4 | 0.4 | 0.1–4.9 |
| **Dichotomized IRTmax[a]** | | | |
| **IRTmax + CS + farm** | IRTmax < 31.0˚C | 1 | referent |
| | IRTmax ≥ 31.0˚C | 13.9 | 3.4–95.7 |
| | CS fresh manure | 1 | referent |
| | CS dried manure | 4.0 | 1.6–10.8 |
| | Farm 1 | 1 | referent |
| | Farm 2 | 2.2 | 0.5–15.2 |
| | Farm 3 | 6.4 | 1.3–48.7 |
| | Farm 4 | 0.5 | 0.1–5.6 |
| **Washed hind feet** | | | |
| **Continuous IRTmax** | | | |
| **IRTmax + CS + farm** | IRTmax | 1.4 | 1.1–1.7 |
| | CS fresh manure | 1 | referent |
| | CS dried manure | 5.3 | 2.2–14.1 |
| | Farm 1 | 1 | referent |
| | Farm 2 | 3.9 | 1.1–17.9 |
| | Farm 3 | 10.7 | 2.7–56.0 |
| | Farm 4 | 1.9 | 0.3–11.4 |
| **Dichotomized IRTmax[a]** | | | |
| **IRTmax + CS + farm** | IRTmax < 31.3˚C | 1 | referent |
| | IRTmax ≥ 31.3˚C | 4.8 | 1.7–15.8 |
| | CS fresh manure | 1 | referent |
| | CS dried manure | 5.5 | 2.3–14.5 |
| | Farm 1 | 1 | referent |
| | Farm 2 | 4.0 | 1.2–18.5 |
| | Farm 3 | 9.4 | 2.4–48.4 |
| | Farm 4 | 2.0 | 0.3–11.4 |

[a] IRTmax was divided into 2 groups over the median value of IRTmax, regardless of M-score [17], in the datasets of unwashed and washed feet, respectively.

In analogy with machine learning techniques used for automated mastitis or oestrus detection [30, 31], similar techniques can be developed to use IRTmax for the detection of M2 lesions in which the IRTmax from a foot is compared with rolling averages of the same foot, contralateral foot, feet average within cow, herd average, or a combination of these. To date, the authors are unaware of publications that report investigations of this option.

A limitation of this study was the low prevalence of feet with M2 lesions and of lame feet. Although this is a realistic reflection of the average Canadian dairy herd [27, 32], it resulted

**Table 4. Final reduced multivariable logistic regression analyses to test the association between maximum infrared temperature (IRTmax) from the plantar pastern and presence of any lesions of digital dermatitis on hind feet from standing dairy cattle before and after washing, with farm as fixed effect.**

| Model | Variable | Adjusted OR | 95% CI |
|---|---|---|---|
| **Unwashed hind feet** | | | |
| **Continuous IRTmax** | | | |
| **IRTmax + farm** | IRTmax | 1.1 | 1.1–1.2 |
| | Farm 1 | 1 | referent |
| | Farm 2 | 1.4 | 0.7–2.8 |
| | Farm 3 | 0.7 | 0.3–1.4 |
| | Farm 4 | 1.2 | 0.6–2.8 |
| | Farm 5 | 0.6 | 0.3–0.9 |
| **Dichotomized IRTmax[a]** | | | |
| **IRTmax + farm** | IRTmax < 31.0˚C | 1 | referent |
| | IRTmax ≥ 31.0˚C | 1.4 | 0.9–2.1 |
| | Farm 1 | 1 | referent |
| | Farm 2 | 1.6 | 0.8–3.2 |
| | Farm 3 | 0.6 | 0.3–1.2 |
| | Farm 4 | 1.3 | 0.6–2.9 |
| | Farm 5 | 0.6 | 0.3–0.9 |
| **Washed hind feet** | | | |
| **Continuous IRTmax** | | | |
| **IRTmax + farm** | IRTmax | 1.1 | 1.1–1.2 |
| | Farm 1 | 1 | referent |
| | Farm 2 | 2.7 | 1.4–5.2 |
| | Farm 3 | 1.3 | 0.7–2.7 |
| | Farm 4 | 2.4 | 1.1–5.4 |
| | Farm 5 | 0.8 | 0.5–1.4 |
| **Dichotomized IRTmax[a]** | | | |
| **IRTmax + farm** | IRTmax < 31.3˚C | 1 | referent |
| | IRTmax ≥ 31.3˚C | 1.2 | 0.8–1.7 |
| | Farm 1 | 1 | referent |
| | Farm 2 | 2.6 | 1.4–5.1 |
| | Farm 3 | 0.9 | 0.5–1.7 |
| | Farm 4 | 2.1 | 1.0–4.7 |
| | Farm 5 | 0.7 | 0.4–1.2 |

[a] IRTmax was divided into two groups over the median value of IRT max, regardless of M-score [17], in the datasets of unwashed and washed feet, respectively.

in a statistically unbalanced dataset. It is possible that this restricted the capacity of our study to detect an association with IRTmax. Also, our dataset contained a large number of animals with only one observation, making the inclusion of cow as a random effect, to account for repeated measures and cows having more than one observation, in our models impossible.

The M-score of the feet in this study was determined by visual inspection of the feet in the milking parlour. Although visual detection of M-scores in the milking parlour versus in the trimming chute was validated [7], we hereby compared IRTmax with an imperfect diagnostic test. Potentially, IRTmax could have correctly diagnosed some feet with M2 lesions that were misclassified as feet without M2 lesions by the in-parlour M-scoring due to the limited Se (0.62) of in-parlour M-scoring for M2 lesions [7]. Diagnosis of DD

in the trimming chute would have reduced possible misclassification of M-scores. Additionally, information on the presence of other foot lesions, such as claw horn lesions, could have been collected as feet with other foot lesions typically tend to have higher IRTmax values compared to feet with no lesions [12, 33–35]. However, inspection of feet in a trimming chute would have neglected the need for an easy, practical method. Higher IRTmax values from cattle feet have also been associated with higher ambient temperatures [33, 34], stage of lactation ≤ 200 DIM [33], and more recently with higher locomotion scores in a herd without DD [36]. Some of these factors will have been captured by fixing farm into the models, but it is likely that they exert an unmeasured effect on the results of our study.

Further research should aim to include all above-mentioned factors with a preference for longitudinal studies to better evaluate IRT as an early detection method for M2 lesions resulting in lameness. However, these multiple factors which influence the ability to detect M2 lesions, and foot lesions in general, all need to be automatically measured and considered before IRT can be easily used as a detection tool on farm. Until this further research is done, the main potential use of IRT in automated detection of foot health status is likely limited to identify 'feet at risk' that need further attention. At-risk feet could either be visually appraised in the trimming chute, or by computer vision and machine learning technology. The YOLOv2 computer vision model of Cernek et al. [37] correctly classified about 60% of the lesions as an M2 lesion on washed hind feet in an external validation trial on a commercial US dairy herd. Combining IRT with other automated lameness detection devices presumably aids in the identification of feet at risk of compromised foot health.

## Conclusions

The presence of M2 lesions on hind feet was associated with higher IRTmax values of the plantar pastern, both on unwashed and washed feet from standing dairy cattle. Dichotomization of IRTmax substantially decreased the reliability of this association, making it unlikely that IRTmax alone can be used for automated detection of feet with an M2 lesion. It is probable that IRTmax does have a role in identifying feet at-risk for compromised foot health that need further checking and thereby is a tool that can aid in the automation of monitoring the foot health status on dairy herds.

## Supporting information

**S1 Table. Technical data for the FLiR i3 handheld infrared thermography camera.**
(DOCX)

**S1 File. Regression analyses M2 lesions.** Full results of all regression analyses performed investigating the association between maximum infrared temperature (IRTmax) of the plantar pastern region of feet from standing dairy cattle and the presence of M2 lesions of digital dermatitis.
(DOCX)

**S2 File. Regression analyses digital dermatitis lesions.** Full results of all regression analyses performed investigating the association between maximum infrared temperature (IRTmax) of the plantar pastern region of feet from standing dairy cattle and the presence of any lesions of digital dermatitis.
(DOCX)

## Acknowledgments

The authors thank participating farmers for their willingness, time, and cooperation on this project. Additional thanks to Kelsey Gray and Gwen Roy (University of Calgary, Alberta, Canada) for assistance with data collection. Special thank you to Al Schaefer and Pierre Lepage (Lacombe Research and Development Centre, Alberta, Canada) for assistance and guidance regarding the use of IRT cameras. The authors are also grateful for the advice of Hans Vernooij (Utrecht University, Utrecht, the Netherlands) on the statistical analyses and the help of Lisanne van der Voort (Utrecht University, Utrecht, the Netherlands) for preparing the figures.

## Author Contributions

**Conceptualization:** Arne Vanhoudt, Casey Jacobs, Herman W. Barkema, Mirjam Nielen, Tine van Werven, Karin Orsel.

**Data curation:** Casey Jacobs, Maaike Caron.

**Formal analysis:** Arne Vanhoudt, Casey Jacobs, Maaike Caron.

**Funding acquisition:** Casey Jacobs, Herman W. Barkema, Karin Orsel.

**Investigation:** Arne Vanhoudt, Casey Jacobs, Maaike Caron, Mirjam Nielen, Tine van Werven, Karin Orsel.

**Methodology:** Arne Vanhoudt, Casey Jacobs, Mirjam Nielen, Tine van Werven, Karin Orsel.

**Project administration:** Casey Jacobs, Karin Orsel.

**Resources:** Casey Jacobs, Herman W. Barkema, Mirjam Nielen, Tine van Werven, Karin Orsel.

**Software:** Arne Vanhoudt, Casey Jacobs, Maaike Caron.

**Supervision:** Casey Jacobs, Herman W. Barkema, Mirjam Nielen, Tine van Werven, Karin Orsel.

**Validation:** Arne Vanhoudt, Casey Jacobs, Herman W. Barkema, Mirjam Nielen, Tine van Werven, Karin Orsel.

**Visualization:** Arne Vanhoudt, Casey Jacobs, Maaike Caron.

**Writing – original draft:** Arne Vanhoudt, Casey Jacobs, Maaike Caron.

**Writing – review & editing:** Casey Jacobs, Herman W. Barkema, Mirjam Nielen, Tine van Werven, Karin Orsel.

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
