## [Decision Letter · Decision Letter 0]

19 Jul 2022

PONE-D-22-12645Broad-spectrum infrared thermography for detection of M2 digital dermatitis lesions on hind feet of standing dairy cattlePLOS ONE

Dear Dr. Vanhoudt,

Thank you for submitting your manuscript to PLOS ONE. After careful consideration, we feel that it has merit but does not fully meet PLOS ONE’s publication criteria as it currently stands. Therefore, we invite you to submit a revised version of the manuscript that addresses the points raised during the review process.

Dear Sir,

Please attend to the reviewer's suggestions to go ahead with the procedures.

Best regards,==============================

We look forward to receiving your revised manuscript.

Kind regards,

Julio Cesar de Souza, Ph.D.

Academic Editor

PLOS ONE

Journal Requirements:

2. In your Methods section, please provide additional details regarding participant consent from the owners of the animals. In the ethics statement in the Methods and online submission information, please ensure that you have specified (1) whether consent was informed and (2) what type you obtained (for instance, written or verbal). If the need for consent was waived by the ethics committee, please include this information.

3. PLOS journals require authors to make all data underlying the findings described in their manuscript fully available without restriction unless the data are subject to ethical restrictions or owned by someone other than the authors (https://journals.plos.org/plosone/s/data-availability#loc-acceptable-data-access-restrictions). Therefore, we ask that you please upload underlying data to an appropriate data repository and update your Data Availability Statement accordingly or provide all contact details for where an interested researcher would need to apply to gain access to the relevant data. Please note that it is not acceptable for an author to be the sole named individual responsible for ensuring data access

“This study was funded by CAAP (Canadian Agriculture Adaptation Program; Canada), Alberta Milk (Edmonton, Alberta, Canada), and DeLaval Manufacturing (Kansas City, Missouri, United States of America). Additional thanks to Kelsey Gray and Gwen Roy (University of Calgary, Alberta, Canada) for assistance with data collection. Special thank you to Al Schaefer and Pierre Lepage (Lacombe Research and Development Centre, Alberta, Canada) for assistance and guidance regarding the use of IRT cameras and for allowing the use of the FLiR i3 camera. The authors are also grateful for the advice of Hans Vernooij (Utrecht University, Utrecht, the Netherlands) on the statistical analyses and the help of Lisanne van der Voort (Utrecht University, Utrecht, the Netherlands) for preparing the figures.”

“This study was funded by CAAP (Canadian Agriculture Adaptation Program; Canada; http://omaf.gov.on.ca/english/food/industry/can-agri-adapt.htm), Alberta Milk (Edmonton, Alberta, Canada; https://albertamilk.com/), and DeLaval Manufacturing (Kansas City, Missouri, United States of America; https://www.delaval.com/en-us/)

Additional Editor Comments:

Dear Sir,

please attend to the reviewer's suggestions to go ahead with the procedures.

Best regards,

Reviewers' comments:

Reviewer's Responses to Questions

**Comments to the Author**

1. Is the manuscript technically sound, and do the data support the conclusions?

Reviewer #1: Yes

Reviewer #2: Yes

2. Has the statistical analysis been performed appropriately and rigorously? 

Reviewer #1: Yes

Reviewer #2: I Don't Know

3. Have the authors made all data underlying the findings in their manuscript fully available?

Reviewer #1: Yes

Reviewer #2: Yes

4. Is the manuscript presented in an intelligible fashion and written in standard English?

Reviewer #1: Yes

Reviewer #2: Yes

5. Review Comments to the Author

Reviewer #1: Introduction: readable, comprehensive, and covering the subject quite right. The first paragraph in the introduction should be based on a reference. Results: good and clear. The titles of the tables in the results (Tables 1, 2, 3 and 4) are too long. I suggest that they be abbreviated.

Discussion: The results needs more discussion. The authors should mention the reasons for the results in detail.

Figure 1, 2 and 3 are not clear and need more accuracy.

Reviewer #2: With interest I read your paper. The paper provides relevant data regarding using IRT to detect M2 lesions in dairy cattle. Well written!

Below you find my minor comments

53-55: add appropriate reference(s) and the same for lines 57-58.

95: have you investigates the difference between ulcerative and proliferative M2 lesions (according to ICAR guidelines in cattle). I expected some differences.

125-126: warm/cold water source?

137: see please my previous comment (ulcerative/ proliferative M2?)

Have you measure the room temperature/humidity. This would higly influce your results as you measure the absolute values of IRTmax rather than the difference. Are all measurement done within optimal condition? When you applied the IRT-Imaging after the washing? Directly or in due of time?

207: observation done as well for fore limbs? Please add details to claw trimming performed on farms. Any anamnestic data are available and already treatment procedures of DD on farms.

Effect of other foot lesions? Either skin or horn lesion must be reported. How other lesions may affect your outcome?

6. PLOS authors have the option to publish the peer review history of their article (what does this mean?). If published, this will include your full peer review and any attached files.

Reviewer #1: **Yes: **Prof. Dr. Khalid Chillab Kridie Al-Salhie

Reviewer #2: No

---

## [Author Response · Author response to Decision Letter 0]

7 Dec 2022

PONE-D-22-12645

Broad-spectrum infrared thermography for detection of M2 digital dermatitis lesions on hind feet of standing dairy cattle

AU: Apologies, we checked the templates and hope we now meet the PLOS ONE’s style requirements.

2. In your Methods section, please provide additional details regarding participant consent from the owners of the animals. In the ethics statement in the Methods and online submission information, please ensure that you have specified (1) whether consent was informed and (2) what type you obtained (for instance, written or verbal). If the need for consent was waived by the ethics committee, please include this information.

AU: We obtained written informed consent from the farmers who participated in this study (L108).

3. PLOS journals require authors to make all data underlying the findings described in their manuscript fully available without restriction unless the data are subject to ethical restrictions or owned by someone other than the authors (https://journals.plos.org/plosone/s/data-availability#loc-acceptable-data-access-restrictions). Therefore, we ask that you please upload underlying data to an appropriate data repository and update your Data Availability Statement accordingly or provide all contact details for where an interested researcher would need to apply to gain access to the relevant data. Please note that it is not acceptable for an author to be the sole named individual responsible for ensuring data access.

 AU: Due to a signed agreement between the researchers and farmers we are unable to make all data underlying the findings of this manuscript fully available without restriction. Access to the anonymised dataset for research involving the research group that collected the original data is possible, and enquiries for this purpose can be made by contacting the corresponding author or research group (headpah@ucalgary.ca or karin.orsel@ucalgary.ca).

“This study was funded by CAAP (Canadian Agriculture Adaptation Program; Canada), Alberta Milk (Edmonton, Alberta, Canada), and DeLaval Manufacturing (Kansas City, Missouri, United States of America). Additional thanks to Kelsey Gray and Gwen Roy (University of Calgary, Alberta, Canada) for assistance with data collection. Special thank you to Al Schaefer and Pierre Lepage (Lacombe Research and Development Centre, Alberta, Canada) for assistance and guidance regarding the use of IRT cameras and for allowing the use of the FLiR i3 camera. The authors are also grateful for the advice of Hans Vernooij (Utrecht University, Utrecht, the Netherlands) on the statistical analyses and the help of Lisanne van der Voort (Utrecht University, Utrecht, the Netherlands) for preparing the figures.”

“This study was funded by CAAP (Canadian Agriculture Adaptation Program; Canada; http://omaf.gov.on.ca/english/food/industry/can-agri-adapt.htm), Alberta Milk (Edmonton, Alberta, Canada; https://albertamilk.com/), and DeLaval Manufacturing (Kansas City, Missouri, United States of America; https://www.delaval.com/en-us/)

AU: Thank you for this guidance. We deleted funding related statements from the Acknowledgements section and provided an amended funding statement in the cover letter.

AU: We have checked all the references cited in this manuscript and made no changes.

Additional Editor Comments:

Dear Sir,

please attend to the reviewer's suggestions to go ahead with the procedures.

Best regards,

Reviewers' comments:

Reviewer's Responses to Questions

Comments to the Author

1. Is the manuscript technically sound, and do the data support the conclusions?

Reviewer #1: Yes

Reviewer #2: Yes

2. Has the statistical analysis been performed appropriately and rigorously? 

Reviewer #1: Yes

Reviewer #2: I Don't Know

3. Have the authors made all data underlying the findings in their manuscript fully available?

Reviewer #1: Yes

Reviewer #2: Yes

4. Is the manuscript presented in an intelligible fashion and written in standard English?

Reviewer #1: Yes

Reviewer #2: Yes

5. Review Comments to the Author

Reviewer #1: 

AU: Thank you for your time and effort reviewing our manuscript. We appreciate your feedback and have done our best to address your comments as good as possible. Line numbers refer to those in the revised manuscript.

Introduction: readable, comprehensive, and covering the subject quite right. The first paragraph in the introduction should be based on a reference. 

AU: We have added references to the first paragraph of the introduction (L56 and 58) .

Results: good and clear. The titles of the tables in the results (Tables 1, 2, 3 and 4) are too long. I suggest that they be abbreviated.

AU: Thanks. We have condensed the titles of the tables a much as possible while adhering to our goal to make the tables interpretable without the manuscript.

Discussion: The results needs more discussion. The authors should mention the reasons for the results in detail.

AU: Can you please provide more specific guidance on what you exactly mean with your request for more discussion?

Currently the discussion includes a summary of the key findings of the study and how these relate to previous work, an outlook for future potential of IRT in DD and foot health control and limitations of the study.

Figure 1, 2 and 3 are not clear and need more accuracy.

AU: We uploaded figure files with higher accuracy.

Reviewer #2: 

AU: Thank you for your time and effort reviewing our manuscript. We appreciate your feedback and have done our best to address your comments as good as possible. Line numbers refer to those in the revised manuscript.

With interest I read your paper. The paper provides relevant data regarding using IRT to detect M2 lesions in dairy cattle. Well written!

Below you find my minor comments

53-55: add appropriate reference(s) and the same for lines 57-58.

AU: We added references for these statements (L56 and 58)

95: have you investigates the difference between ulcerative and proliferative M2 lesions (according to ICAR guidelines in cattle). I expected some differences.

AU: Thank you for this interesting hypothesis. In this study we did not differentiate between ulcerative and proliferative lesions. The data for this study was collected before ulcerative proliferative was included in ICAR guideline. Retrospectively we recall most M2 lesions were ulcerative, but this level of detail was not recorded.

125-126: warm/cold water source?

AU: Water temperature was not recorded. We used a water source as was available in the parlour. L132

137: see please my previous comment (ulcerative/ proliferative M2?)

AU: The data for this study was collected before ulcerative proliferative was included in ICAR guideline.

Have you measured the room temperature/humidity. 

This would highly influence your results as you measure the absolute values of IRTmax rather than the difference. 

AU: The IRT camera was calibrated at the start of each visit (L168-171).

Are all measurement done within optimal condition? 

AU: With our study we aimed to investigate the on farm application of IRT for the detection of M2 lesions (L95-97), hence conditions were not always optimal but realistic.

When you applied the IRT-Imaging after the washing? Directly or in due of time?

AU: Thank you for pointing out that this is not clear. We moved the sentence on the timing of the IRT images of washed feet to L133-135 and added additional information. We hope this is clear now.

“For the IRT images of washed feet, the amount of time between washing feet and capturing the second IRT image varied according to the milking routine and size of the milking parlour.” 

207: observation done as well for fore limbs? 

AU: We only studied hind feet (L96, 129 and 156)

Please add details to claw trimming performed on farms. Any anamnestic data are available and already treatment procedures of DD on farms.

AU: Details on the claw trimming protocols on the farms were not available. All farms continued their regular treatment and hoof trimming schedule during the entire study duration. The farms participated in a footbathing study (Jacobs et al. 2017). Information added on L120-124. 

Effect of other foot lesions? Either skin or horn lesion must be reported. 

How other lesions may affect your outcome?

AU: For the primary aim of this study , i.e. determine whether broad spectrum IRT from unwashed hind feet of cows standing in a milking parlour was associated with M2 lesions, we needed to test the IRT performance without prior information on presence or absence of other lesions. We discuss the effect of other lesions on IRT recordings and our results on L320-324.

6. PLOS authors have the option to publish the peer review history of their article (what does this mean?). If published, this will include your full peer review and any attached files.

Do you want your identity to be public for this peer review? For information about this choice, including consent withdrawal, please see our Privacy Policy.

Reviewer #1: Yes: Prof. Dr. Khalid Chillab Kridie Al-Salhie

Reviewer #2: No

---

## [Editor Report · Decision Letter 1]

20 Dec 2022

Broad-spectrum infrared thermography for detection of M2 digital dermatitis lesions on hind feet of standing dairy cattle

PONE-D-22-12645R1

Dear Dr. Vanhoudt,

We’re pleased to inform you that your manuscript has been judged scientifically suitable for publication and will be formally accepted for publication once it meets all outstanding technical requirements.

Kind regards,

Julio Cesar de Souza, Ph.D.

Academic Editor

PLOS ONE

Additional Editor Comments (optional):

Accept
---

## [Editor Report · Acceptance letter]

4 Jan 2023

PONE-D-22-12645R1 

Broad-spectrum infrared thermography for detection of M2 digital dermatitis lesions on hind feet of standing dairy cattle 

Dear Dr. Vanhoudt:

I'm pleased to inform you that your manuscript has been deemed suitable for publication in PLOS ONE. Congratulations! Your manuscript is now with our production department. 

Kind regards, 

on behalf of

Dr. Julio Cesar de Souza 

Academic Editor

PLOS ONE